# Bypassing the Kochen–Specker Theorem: An Explicit Non-Contextual Statistical Model for the Qutrit

**David H. Oaknin** 

Rafael Ltd., IL-31021 Haifa, Israel; d1306av@gmail.com

**Abstract:** We describe an explicitly non-contextual statistical model of hidden variables for the qutrit, which fully reproduces the predictions of quantum mechanics, and thus, bypasses the constraints imposed by the Kochen–Specker theorem and its subsequent reformulations. We notice that these renowned theorems crucially rely on the implicitly assumed existence of an absolute frame of reference with respect to which physically indistinguishable tests related by spurious gauge transformations can supposedly be assigned well-defined distinct identities. We observe that the existence of such an absolute frame of reference is not required by fundamental physical principles, and hence, assuming it is an unnecessarily restrictive demand.

**Keywords:** quantum mechanics; Kochen–Specker theorem; Bell's theorem; hidden variables; non-contextual models; counterfactual measurements; gauge symmetries; holonomy; statistical physics

**MSC:** Primary 81P13; Secondary 81P10; 81P20

## 1. Introduction

Quantum mechanics is widely regarded as the ultimate mathematical framework within which a hypothetical final theory of Nature's basic building blocks and their interactions might be formulated. This idea is strongly rooted in the fact that quantum phenomena cannot be fully described within any model of underlying hidden variables that shares certain physically intuitive features, as firmly stated by the renowned Bell and Kochen–Specker theorems [1,2] and their subsequent variations and reformulations [3–10].

Indeed, quantum mechanics has been successfully applied to describe an extremely wide range of phenomena in particle physics, cosmology, astrophysics, condensed matter physics, nuclear physics, atomic and molecular physics, and optics. The only remarkable exception in this highly successful program is Einstein's general relativity theory of gravitation for which there does not yet exist even an experimentally testable quantum mechanical candidate in spite of the great efforts invested during the last fifty years [11,12].

On the other hand, fundamental questions raised more than eighty years ago by Einstein, Podolsky, and Rosen through their renowned EPR paradox [13] about the interpretation and completeness of the quantum formalism and the role played by measurements, still remain open. The lack of a clear answer to these fundamental questions points necessarily to an insufficient understanding of all the assumptions involved, either explicitly or implicitly, in the proof of the renowned theorems cited above and lay out the possibility that quantum mechanics could indeed be no more than an emerging framework that provides only an incomplete description of a yet unknown underlying deeper reality [14].

In fact, in a series of recent papers [15–17] we have shown that Bell's theorem, both its original version, as well as its later reformulations and variations, crucially rely on an implicit assumption that had gone unnoticed, namely, the existence of an absolute frame of reference with respect to which the hypothetical hidden configurations of the entangled quantum state, as well as the setting of the detectors that test them, can be described. The existence of an absolute frame allows assigning well-defined distinct identities to

measurement settings that would be symmetrically indistinguishable otherwise. The existence of such an absolute frame of reference, however, is not required by fundamental physical principles and it is, therefore, an assumption that might not be fulfilled in the actual experiments that test the consequences of these theorems. One may think, as a simple but illustrative example, about a spherical surface and a pair of vectors tangent to it at two different locations. Due to the holonomy of the sphere, the relative angle between the two vectors is properly defined only after drawing a path on the surface to connect them. For the same reason, the orientation of either one of the vectors cannot be defined independently from the other.

We further showed in those papers [15–17] that in the absence of an absolute frame of reference it is straightforward to explicitly build a statistical local model of hidden variables that reproduces the predictions of quantum mechanics for the Bell states. In this paper we focus on the Kochen–Specker theorem [2] and, in particular, on the reformulation discussed by Klyachko, Can, Biniciglu and Shumovsky in [7] and show that also these theorems rely on the same disputed assumption. Namely, these theorems also assume the existence of an absolute frame of reference with the help of which distinct identities are assigned to otherwise symmetrically indistinguishable measurements. We follow this insight to build an explicitly non-contextual model of hidden variables for the qutrit by dropping this unnecessary restrictive assumption.

It may be worth reminding the reader at this point that while the Bell theorem tests the apparent non-local features of quantum mechanics, the Kochen–Specker theorem highlights its apparent contextuality. Thus, the Bell theorem involves causally separated measurement events performed on two subsystems, while the Kochen–Specker theorem may involve only measurements performed on a single system, e.g., a qutrit. Furthermore, while the violation in quantum mechanics of the constraints imposed by the Bell theorem requires the entanglement of the two separated subsystems, the violation of the constraints imposed by the Kochen–Specker theorem does not necessarily involve entanglement.

The paper is organized as follows. In Section 2 we review the Kochen–Specker theorem and its subsequent reformulations. We focus in particular on the Klyachko-Can-Biniciglu-Shumovsky (KCBS) inequality [7], which is the simplest statement of the impossibility to reproduce the predictions of quantum mechanics for the qutrit within the framework of non-contextual models of hidden variables that share certain intuitive features. In Section 3 we discuss how the disputed implicit assumption mentioned above appears in the proof of these theorems. In Section 4 we revisit the predictions of quantum mechanics for the qutrit and in Section 5 we build an explicit non-contextual model of hidden variables that reproduces these predictions. Needless to say that the model that we present does not fulfill the disputed assumption. In Section 6 we summarize our conclusions.

## 2. The KCBS Version of the Kochen–Specker Theorem

The Kochen–Specker theorem [2] is one of the pillars upon which relies the widely accepted claim about the impossibility to accommodate the Einstein–Podolsky–Rosen notion of physical realism within the framework of quantum mechanics. The theorem establishes a logical contradiction between the predictions of quantum mechanics and those of generic non-contextual models of hidden variables that share certain apparently trivial intuitive features. Namely, the Kochen–Specker theorem states the impossibility to assign values, either $+1$ or $-1$, to some family of binary tests $\{\mathcal{T}_1, \mathcal{T}_2, \ldots, \mathcal{T}_m\}$ in a way consistent with the predictions of quantum mechanics for whatever state in a Hilbert space of linear dimension larger than two.

Subsequent reformulations of the theorem reach similar conclusions with the help of reduced families of binary tests by making their scope more specific, while keeping the essential assumptions of the original formulation. The simplest of these reformulations is the Klyachko–Can–Biniciglu–Shumovsky (KCBS) theorem for the qutrit [7], which leads to an inequality that holds for any generic non-contextual model of hidden variables that

shares certain physically intuitive features but quantum mechanics violates it. The KCBS inequality is obtained as follows(see Figure 1).

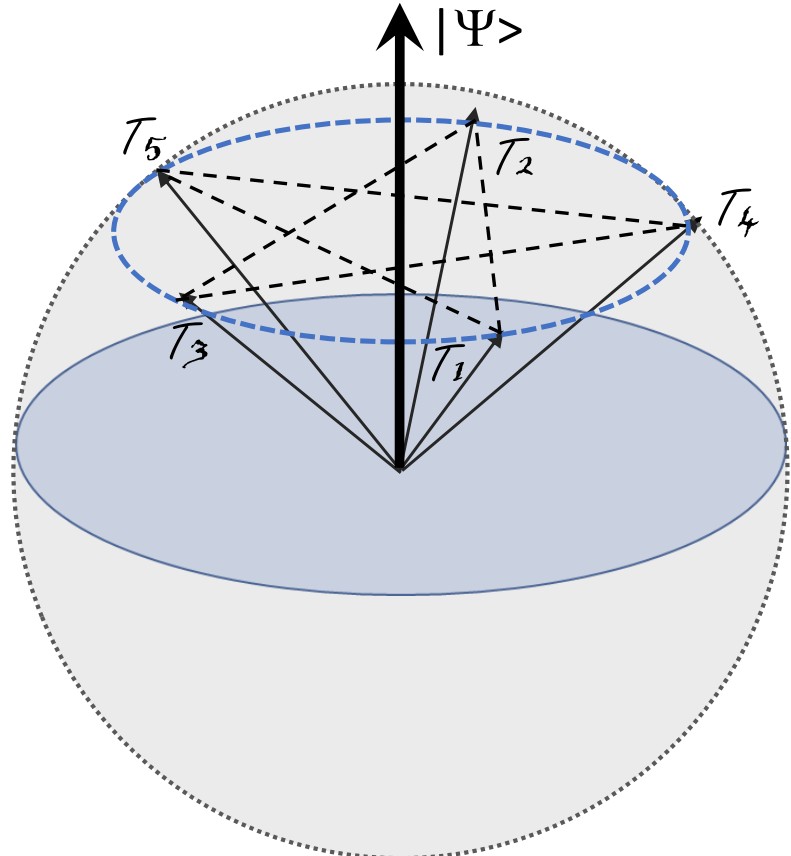

**Figure 1.** The five binary tests $\{\mathcal{T}_i\}_{i=1,2,3,4,5}$ considered by the KCBS theorem relative to the quantum state of the qutrit described by the wavefunction $|\Psi\rangle$. Each pair of consecutive tests $\mathcal{T}_i, \mathcal{T}_{i+1}, i \in \mathbb{N}_{\text{mod}(5)}$ defines a complete set of commuting observables, that is, a *context*. We argue in this paper that all these five *contexts* are related by a gauge transformation and are, therefore, physically indistinguishable.

For any quantum state of the qutrit, which we denote without any loss of generality as

$$|\Psi\rangle = (0,0,1)^t, \tag{1}$$

there seems to exist a set of five binary tests $\{\mathcal{T}_i\}_{i=1,2,3,4,5}$, defined as $\mathcal{T}_i = 1 - 2J_i$, for $J_i = |\chi_i\rangle\langle\chi_i|$ and

$$
\begin{aligned}
|\chi_1\rangle &= \tfrac{1}{\sqrt{1+\cos(\zeta)}} & (1, && 0, && \sqrt{\cos(\zeta)})^t, \\
|\chi_2\rangle &= \tfrac{1}{\sqrt{1+\cos(\zeta)}} & (\cos(4\zeta), && \sin(4\zeta), && \sqrt{\cos(\zeta)})^t, \\
|\chi_3\rangle &= \tfrac{1}{\sqrt{1+\cos(\zeta)}} & (\cos(2\zeta), & - & \sin(2\zeta), && \sqrt{\cos(\zeta)})^t, \\
|\chi_4\rangle &= \tfrac{1}{\sqrt{1+\cos(\zeta)}} & (\cos(2\zeta), && \sin(2\zeta), && \sqrt{\cos(\zeta)})^t, \\
|\chi_5\rangle &= \tfrac{1}{\sqrt{1+\cos(\zeta)}} & (\cos(4\zeta), & - & \sin(4\zeta), && \sqrt{\cos(\zeta)})^t,
\end{aligned}
\tag{2}
$$

with $\zeta = \pi/5$, such that

$$J_i \cdot J_{i+1} = J_{i+1} \cdot J_i = 0, \tag{3}$$

and hence,

$$
\begin{aligned}
p_{[\mathcal{T}_i=-1,\mathcal{T}_{i+1}=+1]} &= |\langle \chi_i | \Psi \rangle|^2 &= \frac{\cos(\zeta)}{1+\cos(\zeta)}, \\
p_{[\mathcal{T}_i=+1,\mathcal{T}_{i+1}=-1]} &= |\langle \chi_{i+1} | \Psi \rangle|^2 &= \frac{\cos(\zeta)}{1+\cos(\zeta)}, \\
p_{[\mathcal{T}_i=+1,\mathcal{T}_{i+1}=+1]} &= \frac{1-\cos(\zeta)}{1+\cos(\zeta)}, \\
p_{[\mathcal{T}_i=-1,\mathcal{T}_{i+1}=-1]} &= 0,
\end{aligned}
\tag{4}
$$

for $i = 1, 2, 3, 4, 5$ (where the addition $i + 1$ is understood as $(i + 1) \bmod 5$, so that $5 + 1 \equiv 1$). Therefore,

$$
\sum_{i=1}^{5} \langle \mathcal{T}_i \rangle = \sum_{i=1}^{5} \langle \Psi | \mathcal{T}_i | \Psi \rangle = 5 - 2 \sum_{i=1}^{5} |\langle \chi_i | \Psi \rangle|^2 = \tag{5}
$$

$$
= 5 \times \left( 1 - \frac{2\cos(\zeta)}{1+\cos(\zeta)} \right) = 5 \times \frac{1 - \cos(\zeta)}{1 + \cos(\zeta)} = 0.52786 < 1.
$$

Furthermore, it follows from (3) that $\mathcal{T}_i \cdot \mathcal{T}_{i+1} = \mathcal{T}_{i+1} \cdot \mathcal{T}_i = 1 - 2(J_i + J_{i+1})$, which implies

$$
[\mathcal{T}_i, \mathcal{T}_{i+1}] = 0, \tag{6}
$$

that is, any two such tests can be performed on the system without any of them affecting the outcome of the other, and therefore, they define a *context*. Moreover, according to (4), for none of these *contexts*, there exists a common eigenstate that would produce two negative outcomes; at least one of the two compatible tests must produce a positive outcome.

These predictions cannot be reproduced by any statistical model in which each possible hidden configuration, denoted here generically as $\theta \in \mathcal{S}$, is assigned a binary 5-tuple $\{t_i(\theta)\}_{i=1,2,3,4,5} \in \{-1, +1\}^5$ to describe the outcomes that would be obtained for each one of the five considered tests, since these assignments must fulfill that

$$
t_i(\theta) = -1 \Rightarrow ((t_{i+1}(\theta) = +1) \wedge (t_{i-1}(\theta) = +1)), \tag{7}
$$

for $i = 1, 2, 3, 4, 5$, because two consecutive tests must never produce both a negative outcome. Therefore,

$$
\sum_{i=1}^{5} t_i(\theta) \geq 1, \tag{8}
$$

and

$$
\left\langle \sum_{i=1}^{5} t_i(\theta) \right\rangle_{\theta \in \mathcal{S}} \geq 1, \tag{9}
$$

in contradiction with the prediction (5) of quantum mechanics.

## 3. An Unnoticed Implicit Assumption

The proof of the KCBS theorem presented above, as well as the proof of all other versions of the Kochen–Specker theorem, are straightforward and obviously correct. The main claim of this paper, however, is, as we already noticed in the Introduction, that these proofs rely on a crucial implicit assumption that is not required by fundamental physical principles, and therefore, might not be fulfilled in the experiments that test the implications of the theorems. Of course conclusions reached under certain assumed conditions are not necessarily fulfilled when the latter do not hold.

As a simple but instructive simile, let's consider the statement that the distance covered by a point particle free-falling in a constant gravitational field grows quadratically with time if its initial velocity was zero. Even though the proof of this statement is straightforward, the conclusion is not fulfilled when the gravitational field is not constant.

We showed in [15–17] that the proof of Bell's theorem, in all its versions, implicitly assumes that there exists an absolute frame of reference with respect to which can be described the hypothetical hidden configurations of the pair of entangled qubits, as well as the setting of the two detectors that test them. We argued that the existence of such an absolute frame of reference is not required by fundamental physical principles and, in fact, demanding its existence is at odds with these principles. Here we focus on how this same implicit assumption appears also in the proof of the Kochen–Specker theorem and its later reformulations.

Let us first remind the reader that every single realization of the qutrit can be tested along one and only one pair of compatible measurements, that is, within a single *context*. Moreover, as it can be immediately seen from (4), all the five measurements (and hence, also the five possible *contexts*) considered by the KCBS theorem) are statistically indistinguishable from each other. In order to assign them distinct identities, which are crucial in the proof of the theorem, an external frame of reference is needed. Nonetheless, according to fundamental physical principles, the choice of a particular external frame of reference should not play any role in the description of the physical system, and hence, neither the identity of the tested *context*, which is a spurious non-physical degree of freedom.

In order to make this point completely clear let us consider the situation shown in Figure 2. We consider two detectors (whose setting is represented by the red and green arrows) that perform compatible measurements on a qutrit (whose quantum state is represented by the black arrow). Even though the pencil laying beside them plays no role whatsoever in the measurements, if its orientation serves as our external frame of reference, by rotating it we could obtain different identities for the considered measurement *context*. The pictures shown in Figure 3 are intended to remind the reader about the cosmological scenario in which any measurement is performed and that neither the pencil nor the orange band in Figure 2 can be preferred over the other as a legitimate external frame.

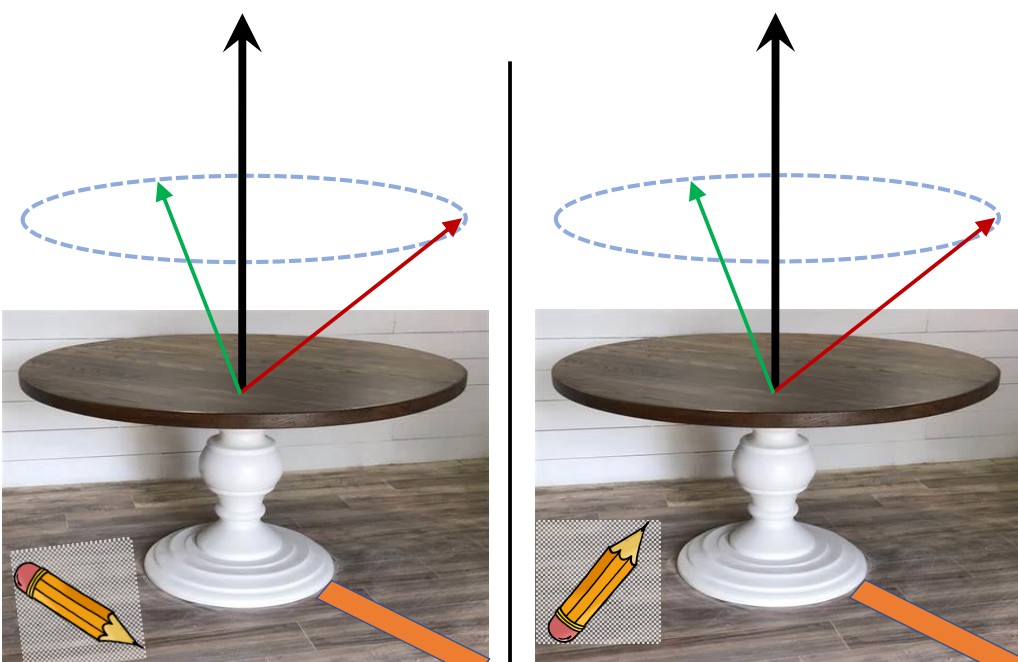

**Figure 2.** Different identities for the *context* defined by two detectors (red and green arrows) performing measurements on a qutrit (black arrow) can be obtained by rotating the pencil that lays beside them when the latter serves as our external frame of reference.

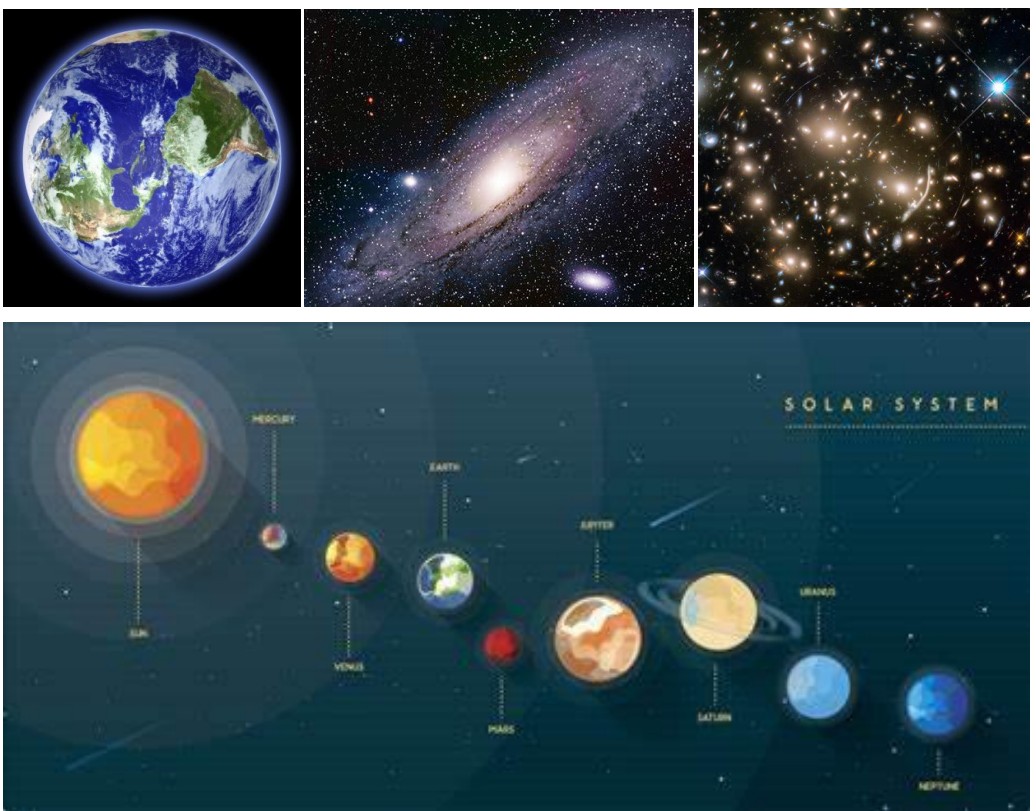

**Figure 3.** In order to avoid any possible misunderstanding about the legitimacy of the pencil shown in Figure 2 to serve as an external frame of reference, we present here a perspective of the cosmological scenario in which all measurements are performed: neither the orange band nor the pencil can be preferred over the other as a legitimate external frame of reference.

A critical reader may counter that this trivial observation is futile since for any single realization of the qutrit five distinct possible counter-factual measurements can be defined with respect to the chosen external frame, whatever this choice is, and therefore, any model of hidden variables must be supposedly capable of assigning definite outcomes for all these five counter-factual measurements, and thus, the conditions required for the theorem to hold are fulfilled. This seemingly obvious counterargument, nevertheless, misses the actual deep implications that the lack of physical identity of the performed measurements has when only two of them, rather than all five, can be performed on every single realization of the qutrit.

In the absence of a well-defined identity for each one of the performed measurements, the hidden configurations of the qutrit can be properly described only with respect to either one of the two measurement devices that actually test them, which we will label in what follows as $A$ and $B$, respectively. By symmetry considerations, these two descriptions must be statistically identical.

In particular, let us denote by $\theta_A \in \mathcal{S}$ the (set of) coordinates that describe how the hidden configuration of the qutrit appears with respect to the measurement device $A$, and by $\theta_B \in \mathcal{S}$ the (set of) coordinates that describe how it appears with respect to the measurement device $B$. Since both (sets of) coordinates describe the same hidden configuration of the qutrit with respect to two different frames of reference (those defined by devices $A$ and $B$, respectively), they must be related by a coordinates transformation:

$$\theta_B = \mathcal{L}(\theta_A; \Theta), \tag{10}$$

where $\Theta$ is a parameter that describes the setting of the two detectors. By symmetry considerations, we posit that the response of the two detectors to the realized hidden

configuration of the qutrit is given by $S(\theta_A)$ and $S(\theta_B)$, respectively, with one and the same response function $S(\cdot)$ for both detectors. By the same symmetry considerations, we require also that if the variable $\theta_A$ is distributed with a density of probability $g(\theta_A)$ over the space of all possible hidden configurations, the variable $\theta_B$ must also be distributed with the same density of probability $g(\theta_B)$. In order to guarantee that the probability to occur of each hidden configuration does not depend on which one of the two descriptions is used, we further demand that

$$d\theta_A \, g(\theta_A) = d\theta_B \, g(\theta_B), \tag{11}$$

which, in turn, demands,

$$g(\theta_A) = \frac{d\mathcal{L}(\theta_A; \Theta)}{d\theta_A} \, g(\mathcal{L}(\theta_A; \Theta)). \tag{12}$$

for any possible values of the parameter $\Theta$.

Similarly, we could also define the (sets of) coordinates

$$\theta_C = \mathcal{L}(\theta_B; \Theta), \quad \theta_D = \mathcal{L}(\theta_C; \Theta), \quad \theta_E = \mathcal{L}(\theta_D; \Theta) \in \mathcal{S}, \tag{13}$$

obtained after consecutive transformations of the (sets of) coordinates. The existence of an absolute frame of reference would enforce that these five (sets of) coordinates would describe how the realized hidden configuration of the qutrit would appear with respect to the five measurement devices considered by the proof of the KCBS theorem, and therefore, we should demand that $\mathcal{L}(\theta_E; \Theta) = \theta_A$. In consequence, the outcomes of the five measurements would be given by the 5-tuple $S(\theta_A), S(\theta_B), S(\theta_C), S(\theta_D), S(\theta_E)$, and the KCBS constraint (9) would hold.

However, in the absence of an absolute frame of reference, we may allow a non-zero geometric phase, $\alpha \neq 0$:

$$\mathcal{L}(\theta_E; \Theta) = \theta'_A = \mathcal{L}(\theta_A; \alpha) \neq \theta_A, \tag{14}$$

which can be accounted for by a spurious redefinition of the identity of the performed measurement. In the presence of a non-zero geometric phase, we cannot consistently define the five measurements considered in the proof of the KCBS theorem and, of course, neither their outcomes. Hence, the conditions required for the theorem to hold are not fulfilled.

## 4. The Quantum Mechanical Predictions For The Qutrit

In this section, we follow our previous observations to build within the framework of quantum mechanics the most general family of counter-factual *contexts* that consist of two compatible binary tests within which we can describe the qutrit.

Without any loss of generality we fix the quantum state of the qutrit being aligned with the Z axis of the local observer's frame of reference (1), and take advantage of the mentioned symmetry to set the first binary test $\mathcal{T}_A \equiv 1 - 2|\xi_A\rangle\langle\xi_A|$ to lay within the XZ plane,

$$|\xi_A\rangle \equiv (\sin\eta, \, 0, \, \cos\eta)^t, \; \eta \in [0, \pi/2], \tag{15}$$

so that its two possible outcomes happen with probabilities

$$\begin{array}{llll} p_{[\mathcal{T}_A=-1]} & = & |\langle\xi_A|\Psi\rangle|^2 & = \cos^2(\eta), \\ p_{[\mathcal{T}_A=+1]} & = & 1 - |\langle\xi_A|\Psi\rangle|^2 & = \sin^2(\eta). \end{array} \tag{16}$$

Any other binary test $\mathcal{T}_B \equiv 1 - 2|\xi_B\rangle\langle\xi_B|$ can then be parameterized as

$$|\xi_B\rangle \equiv (e^{i\mu} \cdot \sin\nu \cdot \cos\omega, \, e^{i\rho} \cdot \sin\nu \cdot \sin\omega, \, \cos\nu)^t,$$

with

$$\nu \in [0, \pi/2), \quad \omega \in [0, \pi/2], \quad \mu, \rho \in [0, 2\pi]. \tag{17}$$

The two tests $\mathcal{T}_A$ and $\mathcal{T}_B$ define a *context* if and only if $\langle \xi_A | \xi_B \rangle = 0$, that is,

$$e^{i\mu} \cdot \sin \nu \cdot \cos \omega \cdot \sin \eta + \cos \nu \cdot \cos \eta = 0 \tag{18}$$

and hence,

$$e^{i\mu} \cdot \tan \nu \cdot \cos \omega \cdot \tan \eta = -1. \tag{19}$$

This condition requires

$$\mu = \pi, \quad \tan \eta \cdot \tan \nu \cdot \cos \omega = 1, \tag{20}$$

and hence, can only be fulfilled for

$$\frac{\pi}{2} - \nu \in [0, \eta]. \tag{21}$$

For each value of $\nu$ in this interval, there exists one and only one value

$$\cos \omega = \frac{1}{\tan \eta \cdot \tan \nu} \tag{22}$$

that solves the orthogonality constraint (18). Therefore,

$$|\xi_B\rangle \equiv \frac{\cos \nu}{\tan \eta} \left( -1, \pm \sqrt{\tan^2 \eta \cdot \tan^2 \nu - 1}, \; \tan \eta \right)^t, \tag{23}$$

where we have arbitrarily set $\rho = 0, \pi$. In summary, the most general *context* $\{\mathcal{T}_A, \mathcal{T}_B\}$ of two binary tests $\mathcal{T}_A \equiv 1 - 2|\xi_A\rangle\langle\xi_A|$, $\mathcal{T}_B \equiv 1 - 2|\xi_B\rangle\langle\xi_B|$) that can be performed on the qutrit (1) can be parameterized by two angles $(\eta, \nu)$ in the range

$$0 \leq \frac{\pi}{2} - \nu \leq \eta \leq \pi/2, \tag{24}$$

such that the first test is defined by eq.(15) and the second test by eq.(23). The probabilities for the possible outcomes for the two compatible tests are given by:

$$\begin{aligned}
p_{[\mathcal{T}_A=-1, \mathcal{T}_B=+1]} &= |\langle \xi_A | \Psi \rangle|^2 = \cos^2(\eta) = \frac{1+\cos(2\eta)}{2}, \\
p_{[\mathcal{T}_A=+1, \mathcal{T}_B=-1]} &= |\langle \xi_B | \Psi \rangle|^2 = \cos^2(\nu) = \frac{1+\cos(2\nu)}{2}, \\
p_{[\mathcal{T}_A=+1, \mathcal{T}_B=+1]} &= -\frac{\cos(2\eta)+\cos(2\nu)}{2}, \\
p_{[\mathcal{T}_A=-1, \mathcal{T}_B=-1]} &= 0.
\end{aligned} \tag{25}$$

These probabilities describe the predictions of quantum mechanics for the most general family of counter-factual *contexts* of compatible binary tests for the qutrit. Any additional geometric structure over-imposed on these predictions is not experimentally testable, and therefore, unnecessarily restrictive.

In particular, for

$$\eta = \nu = \text{arc-cos}\left( \sqrt{\frac{\cos(\zeta)}{1 + \cos(\zeta)}} \right) = 0.83828, \tag{26}$$

we recover the *context* considered in the KBCS theorem: any and all of them!

## 5. Results

A statistical model of hidden variables for the qutrit must be capable to reproduce the predictions (25) that we obtained in the last section, but must not be required to reproduce any additional untestable geometric structure over-imposed on them. The statistical model presented in this section fulfills these demands.

We consider a continuous infinite set of equally probable hidden configurations, each one described by two points $(\hat{r}_1, \hat{r}_2)$ located on the unit sphere. The first point $\hat{r}_1$ defines the principal axis of the qutrit and we can assume without any loss of generality that is directed along the $Z$-axis, as we did in the description of its quantum state (1). The second point $\hat{r}_2$ is randomly distributed over the sphere, and we describe it using the polar angle between the two points, with respect to the center of the sphere. We denote by $\theta_A \in [0, \pi]$ the polar angle as seen from the point of view of test $\mathcal{T}_A$, and by $\theta_B \in [0, \pi]$ the polar angle as seen from the point of view of test $\mathcal{T}_B$. We posit that the two descriptions of the polar angle between $(\hat{r}_1, \hat{r}_2)$ are related by the transformation law

$$\theta_B = \pi - \theta_A, \tag{27}$$

(and hence, also $\theta_A = \pi - \theta_B$). We further posit that the points are uniformly distributed over the sphere so that both random variables are described by the same probability density distribution:

$$g(\theta) = \frac{1}{2}\sin(\theta), \tag{28}$$

so that requirement (11), which states that the probability of each one of the possible hidden configurations to occur does not depend on the (set of) coordinates used to describe it, is fulfilled.

Finally, we define the outcomes for each one of the possible counter-factual *contexts* labeled by the pair of angles $(\eta, \nu)$ in the range (24), in a manifestly symmetric and non-contextual way as follows:

$$\begin{aligned} S_{[\mathcal{T}_A]}(\theta_A) = -1 &\iff \theta_A \in (2\eta, \pi], \\ S_{[\mathcal{T}_A]}(\theta_A) = +1 &\iff \theta_A \in (0, 2\eta], \end{aligned}$$

$$\begin{aligned} S_{[\mathcal{T}_B]}(\theta_B) = -1 &\iff \theta_B \in (2\nu, \pi], \\ S_{[\mathcal{T}_B]}(\theta_B) = +1 &\iff \theta_B \in (0, 2\nu]. \end{aligned}$$

Using the transformation law (27) the response of test $\mathcal{T}_B$ can be written as

$$\begin{aligned} S_{[\mathcal{T}_B]}(\theta_A) = -1 &\iff \theta_A \in (0, \pi - 2\nu], \\ S_{[\mathcal{T}_B]}(\theta_A) = +1 &\iff \theta_A \in (\pi - 2\nu, \pi]. \end{aligned}$$

so that

$$\begin{aligned} p_{[\mathcal{T}_A=-1,\mathcal{T}_B=+1]} &= \frac{1}{2}\int_{2\eta}^{\pi} d\theta \sin(\theta) &= \frac{1+\cos(2\eta)}{2}, \\ p_{[\mathcal{T}_A=+1,\mathcal{T}_B=-1]} &= \frac{1}{2}\int_{0}^{\pi-2\nu} d\theta \sin(\theta) &= \frac{1+\cos(2\nu)}{2}, \\ p_{[\mathcal{T}_A=+1,\mathcal{T}_B=+1]} &= \frac{1}{2}\int_{\pi-2\nu}^{2\eta} d\theta \sin(\theta) &= -\frac{\cos(2\eta)+\cos(2\nu)}{2}, \\ p_{[\mathcal{T}_A=-1,\mathcal{T}_B=-1]} &= 0, \end{aligned}$$

which reproduce the predictions of quantum mechanics (25). As we did already notice above, for $\eta = \nu = \text{arc-cos}\left(\sqrt{\frac{\cos(\zeta)}{1+\cos(\zeta)}}\right) = 0.83828$ we recover the *context* considered in the KCBS theorem, and after applying the transformation law (27) five consecutive times we obtain,

$$\begin{aligned} \theta_C &= \pi - \theta_B = \theta_A, \tag{29} \\ \theta_D &= \pi - \theta_C = \pi - \theta_A, \tag{30} \\ \theta_E &= \pi - \theta_D = \theta_A, \tag{31} \end{aligned}$$

and, finally, a non-zero geometric phase,

$$\theta'_A = \pi - \theta_E = \pi - \theta_A \tag{32}$$

that can, nonetheless, be accounted for by a spurious flip of the orientation of the main axis of the qutrit.

## 6. Discussion

The Kochen–Specker theorem states that non-contextual statistical models of hidden variables that share certain intuitive features cannot reproduce the predictions of quantum mechanics for systems whose Hilbert space of states has a linear dimension larger than $d = 2$ [2], since there always exists a family of binary tests that cannot be assigned specific values, either $+1$ or $-1$, consistent with the quantum predictions without incurring in logical inconsistencies. The simplest quantum system for which the Kochen–Specker theorem applies is the qutrit, $d = 3$, and the simplest version of the above-mentioned logical inconsistency is the KCBS inequality [7].

In this paper, we have shown, however, that the proof of these theorems crucially relies on an implicit assumption that is not required by fundamental physical principles and, hence, it might not be fulfilled in the actual experiments that test the consequences of these theorems. Namely, the proof of the theorems relies on the existence of an absolute frame of reference with respect to which the hidden configuration of the quantum system, as well as the setting of the measurement devices that probe it, can be described. This assumption is based only on physical intuition gained from generic macroscopic systems and, therefore, it should not be taken as a given fact. Once this unjustified assumption is dropped, it is trivial to build an explicitly non-contextual statistical model of hidden variables that reproduces the predictions of quantum mechanics for the qutrit. Thus, the experimentally confirmed violation of the constraints derived under the said assumption may be actually understood as experimental evidence against it.

The conclusions obtained in this paper about the possibility to bypass the constraints imposed by the Kochen–Specker theorem are similar to the conclusions previously reported in [15–17] for Bell's theorem. Our conclusions suggest that quantum mechanics may be an emergent framework that effectively describes an underlying reality obeying still undiscovered principles.

**Funding:** This research received no external funding.

**Institutional Review Board Statement:** Not applicable.

**Informed Consent Statement:** Not applicable.

**Data Availability Statement:** Not applicable.

**Conflicts of Interest:** The authors declare no conflict of interest.

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
