# Peer review of "Bypassing the Kochen–Specker Theorem: An Explicit Non-Contextual Statistical Model for the Qutrit"

_axioms, doi:10.3390/axioms12010090_

Round 1

Reviewer 1 Report

The article claims a possibility of explaining one of the fundamental problems of quantum mechanics, namely the incompatibility between locality and realism, by invoking an ``unjustified hypothesis'' involving the necessity of referring the various instrumental settings to an alleged ``universal reference frame'', the existence of which might be doubted.

While I most certainly will not say that I am convinced, I do not believe my opinion on the matter is relevant. The argument is presented relatively clearly, and readers can surely judge for themselves its validity. On these grounds, the paper might well be published.

My objection to the paper's publication in its present state is of quite a different nature. There have been several forms of the argument showing that quantum mechanics is presumably incompatible with local realism. These involve the CHSH inequalities (generalizing the Bell inequalities), the Greenberger--Horne--Zeilinger experimental arrangement and the Kochen--Specker theorem to name but a few.

The author has already published three papers on related subjects, namely references 14--16 of the paper. All of them present very related ideas, in very similar language. There are additionally various arxiv manuscripts on the same subjects and again with the same ideas: are these, too, going to be published in a series of papers? These are:
1. physics.gen-ph/2008.02633 ``On the roles of Vorob'ev cyclicities and Berry's phase in the EPR paradox and Bell-tests''
2. quant-ph/1411.5704 ``Solving the EPR paradox: an explicit statistical model for the singlet quantum state''
3. arXiv:quant-ph/1709.00167 ``Solving the Greenberger-Horne-Zeilinger paradox: an explicitly local and realistic model of hidden variables for the GHZ quantum state''

So we already have the same essential idea already published in three journals, and three more in a possible pipeline. This is clearly serial publication. Since all these different paradoxes are, in fact, only marginally different, it does not appear that anything significant is gained by such a procedure.

I believe the paper would only be publishable if, for instance, it could show explicitly that all and every ``paradox'' can be solved in the framework provided by the author. This would have the two merits, of showing a result of general interest and of closing the way to further publication of similar papers.

As the paper stands, it should be rejected.

Author Response

Dear Editor,

I would like to thank the referee for his comments and criticisms, which have helped me to improve the paper. 

The referee complains that the ideas contained in this paper have already been discussed in previous publications authored by myself in what he labels as "serial publishing". This criticism is motivated by a misconception, which I have tried to clarify in the revised version of the paper.

While the CHSH-Bell theorem addressed in the previous publications highlights the apparent non-local features of quantum mechanics, the Kochen-Specker theorem addressed in the manuscript under consideration here highlights a different apparently weird feature of quantum mechanics called contextuality.  In fact, while the CHSH-Bell theorem relates to causally disconnected measurement events performed on two separated sub-systems, the Kochen-Specker theorem relates to measurements that may be performed on a single system, for example, a qutrit. Moreover, while the violation of the CHSH-Bell theorem in quantum mechanics demands entangled subsystems, the violation of the Kochen-Specker theorem does not require entanglement. For these and other reasons, the Kochen-Specker theorem is addressed on its own in a separate manuscript. Obviously, it is a strong aspect of the ideas discussed in the manuscript under consideration that both the apparent non-local and contextual features of quantum mechanics may be understood as a result of one and the same unjustified assumption. 

By the way, the third theorem mentioned by the referee - the GHZ theorem - also deserves to be addressed on its own because, unlike the CHSH-Bell theorem, it tests the apparent non-local features of quantum mechanics in a way that does not involve statistical correlations but only single realizations.  

Finally, I wish to remark that I do not share the referee's opinion, according to which all the implications of the ideas discussed in this and previous papers (and others in preparation) should be addressed in a single manuscript. 

Reviewer 2 Report

In this manuscript the author continues his revision, begun with previously published papers, of those works that deals with the fundamental subject of hidden variables and the interpretation of the quantum formalism. After spotting an unnecessary crucial assumption in the Bell theorem for two entangled particles, here he moves on to the theorem of Kochen-Specker and in particular to its simplest reformulation of Klyachko-Can-Binicioglu-Shumovsky for a spin-1 system. Here he finds out that the five squares of the spin projector operators, that appear in the inequality that represents the test for arbitrary hidden variables, are in fact related to each other by gauge transformations and hence are not physically distinguishable, which is conversely crucial in the proof of the KCBS theorem. As he has already done for the Bell theorem, he clarifies that the implicit assumption that wrongly allows to distinguish the squares of the five spin projector operators, i.e. the hidden configurations and the detector settings to test them, is the existence of an absolute reference frame, which is a condition that is not required by fundamental principles of physics. He points out that of the five contexts of measurement only two at a time are actually performed on the spin-1 system. He shows that by dropping the assumption of the absolute frame it is possible to build a non-contextual statistical model of hidden variables whose predictions, unlike the KCBS case, agree with those of quantum, mechanics. In this sense, he concludes that the quantum mechanical formalism may indeed describe an underlying reality not yet discovered.

The paper is clearly written and well balanced. The main aspects of the preceding research are clearly provided along with those features that are further on criticized and reformulated. The point made and the conclusions are sound. The author has spotted an apparently simple flaw in the main and accepted lines of reasoning about the possible existence of hidden variables underlying the formalism of quantum mechanics. In this sense, the paper is very interesting as it sheds a new light on a subject that is still very far from the grasp of physics.

I recommend this paper for publication as it is.

Author Response

Dear Editor,

I wish to warmly thank the referee for his comments.

Reviewer 3 Report

The Bell theorem and its extension given by Kochen and Specker tell us that any local model of underlying hidden variables is unable to describe completely a quantum phenomena. Nevertheless, author claimed the he build an explicitly local model of hidden variables that reproduces the predictions of quantum mechanics  for the Bell states.

The author's main idea is original and relevant to this fundamental  problem of physics. In fact, author's previous works also were devoted to this idea and here he extended this one to for the qutrit quantum states.

The author widely used the notion preferred absolute frame of reference in the relation with experimental setup of the detectors. Author's main idea is that only the relative orientation between the two measurement devices in every single realization of the experiment is a properly defined physical degree of freedom, while their global rigid orientation is a spurious gauge degree of freedom. It should be demonstrated on existed experimental results.

I may suggest  to take into account similar work by
 Giancarlo Ghirardi and Raffaele Romano,
"Onthological models predictively inequivalent to quantum theory".
Physical Review Letters. 110 (17): 170404. arXiv:1301.2695.  
doi:10.1103/PhysRevLett.110.170404.

Author Response

Dear Editor,  

I wish to thank the referee for his comments and the reference he included in his report.  

The referee suggests that the claims made in the paper under consideration should be supported with experimental evidence for the non-existence of an absolute frame of reference, with respect to which symmetrically identical experimental settings might be given distinct identities.

Even though I think that the experimentally confirmed violation of the constraints imposed by the CHSH-Bell and Kochen-Specker theorems under the disputed assumption may be understood as clear evidence against it, I agree with the referee that obtaining direct evidence of an underlying reality would be very important. I am currently considering some ideas along these lines. In any case, it must be clear that the disputed assumption should not be taken as a granted fact unless direct evidence against it is obtained, because it is not required from fundamental principles.

Round 2

Reviewer 1 Report

The author has only added a couple of sentences to the article. My earlier criticisms stand. The paper should not be published.